# Impacts of Hurricane Disturbance on Water Quality across the Aquatic Continuum of a Blackwater River to Estuary Complex

**Tracey Schafer** [1,2,*], **Nicholas Ward** [3,4], **Paul Julian** [1], **K. Ramesh Reddy** [2]
**and Todd Z. Osborne** [1,2]

1    Whitney Marine Lab for Marine Bioscience, University of Florida, St. Augustine, FL 32080, USA;
     pjulian@ufl.edu (P.J.); osbornet@ufl.edu (T.Z.O.)
2    Wetland Biogeochemistry Laboratory, Soil and Water Sciences Department, University of Florida,
     Gainesville, FL 32611, USA; krr@ufl.edu
3    Marine Sciences Laboratory, Pacific Northwest National Laboratory, Sequim, WA 98382, USA;
     nickdward@gmail.com
4    School of Oceanography, University of Washington, Seattle, WA 98195, USA
*    Correspondence: tschafer25@ufl.edu

**Abstract:** Hurricanes cause landscape-scale disturbances that affect biogeochemical cycling and water quality in coastal ecosystems. During Hurricane Irma's passage through northern Florida, water movements driven by wind velocities up to 105 km h$^{-1}$ caused a salinity peak in an estuary/blackwater river complex. Water quality was monitored across the 15 km site to detect the magnitude and duration of disturbance. Saline water intruded 15 km inland into a freshwater portion of the river that peaked at a salinity of 2 psu. Due to the volume of precipitation from the hurricane, significant runoff of freshwater and dissolved organic matter (DOM) caused a decrease in salinity, dissolved oxygen (DO), and Chlorophyll-*a* concentrations while increasing turbidity and fluorescent dissolved organic matter (fDOM). The disturbance caused rapid changes observed by in-situ water quality monitors over a 3-week period, but some effects persisted for longer periods as shown by 3-month weekly water sampling. This disturbance caused shifts in DOM loading, altered salinity dynamics, and reshaped landscapes due to wind and wave surge both in upland marsh and downstream estuary. Hurricane disturbance temporarily and abruptly alters the aquatic continuum, and observations of system response can help us understand the mechanisms associated with ecosystem resilience and recovery.

**Keywords:** extreme disturbance; hurricane; water quality; estuarine biogeochemistry; dissolved organic matter

## 1. Introduction

As global temperatures continue to rise, tropical cyclone activity in the north Atlantic has concomitantly increased over the past 50 years [1]. These storms have the power to substantially impact coastal ecosystems and influence biogeochemical cycling and water quality [2–4]. During these storm events, high velocity winds transport dissolved and particulate materials and affect water levels and salinity [5]. Intense rainfall associated with these storms increases runoff contributions to local waterways resulting in elevated surface water discharge and causing significant loading of organic matter (OM) and sediment to nearby streams and rivers [6,7]. Long term effects that may be associated with these disturbances can cause shifts in coastal ecosystems that both directly and indirectly affect ecosystem resilience and function. For example, Hurricane Katrina caused

widespread mortality of coastal forests around the US Gulf Coast, damaging a reservoir of carbon that represents 50%–140% of the net annual US forest carbon sink [8]. Furthermore, numerous factors contribute to an ecosystem's recovery and resilience including storm characteristics (angle and impact, intensity, surge, etc.) and in-situ characteristics such as nutrient reserves, microbial dynamics, biotic controls, and ecosystem composition [9,10]. Disturbances in the headwaters of the freshwater-to-marine aquatic continuum can have significant ramifications to downstream portions of these ecosystems [11,12]. Nutrients transported by stormwater runoff and physical effects from storm damage in areas upstream will eventually influence areas downstream along the flow path. Typically, freshwater originates from upland ecosystems and flows downstream into larger order streams, rivers, and estuaries, where biological communities replace one another to maintain levels of energy equilibria throughout the length of the waterway [12]. As biological communities, as well as terrestrial and aquatic inputs, change along the length of a river, OM composition and nutrient abundance evolves [11]. A variety of factors modulate OM and nutrient abundance along the river continuum, with timing and magnitude of precipitation and wind being the primary driver of material inputs from land to river [13,14] and hydrodynamics (e.g., flow rates and turbulence) moderating rates of biological respiration [15]. These dynamics become more complex along the transition from river to estuary, where tides drive frequent variability in water level, water chemistry, and terrestrial-aquatic linkages. Thus, disturbances that impact one part of the aquatic continuum will likely affect the rest of these complex interconnected systems.

The impact of hurricanes on water quality has been observed to be widespread and have a sustained effect on the entire system post event [16,17]. Changes in salinity due to forcing from wind and precipitation can lead to impacts such as osmotic stress and, potentially, the mortality of organisms that cannot effectively osmoregulate [18]. Increased suspended material, along with dissolved organic matter (DOM), can increase turbidity and out-shade vegetation leading to decreases in density of submerged aquatic vegetation (SAV) [19,20]. Large nutrient fluxes can also cause harmful algal blooms that affect vegetative, wildlife, and human health [21–23]. The combination of environmental impacts of a particular storm are not universal and depend largely on the magnitude and storm characteristics such as where rainfall occurs (i.e., over land or water) and wind direction [24,25].

During hurricane Irma (10 September 2017–11 September 2017), high winds, several inches of precipitation, and significant storm surge greatly affected the local environment. Large amounts of rainfall and wind from the storm caused influxes of organic matter into waterways and created shifts in local biogeochemical and ecosystem processes [26]. Effects of previous hurricanes on local water quality has been studied [16]; however, high resolution measurements comparing a wide range of parameters along a salinity gradient during a hurricane has not been well documented. These salinity gradients exist as transition zones for coastal systems that are greatly affected by climate change, sea level rise, and large scale disturbances such as hurricanes [27,28]. Thus, it is critical to understand the effects of disturbance events on these transition zones to determine the resilience of these systems.

The primary goal of this opportunistic study was to determine the overall effects of Hurricane Irma on short-term water quality and biogeochemical cycling along a salinity gradient in north-east Florida (USA) and determine the magnitude and duration of storm induced deviations in water quality from normal (pre-storm) conditions. We hypothesized that large scale disturbances due to extreme events such as hurricanes, cause disruption of normal hydrological and biogeochemical conditions that may lead to a hysteretic response that yields a short-term altered state that would persist for less than a month. The resilience and original state of degradation of the system, as well as the scale of the hurricane, should determine the time in which the ecosystem's altered state persists.

## 2. Materials and Methods

### 2.1. Study Site

Hurricane Irma originated as a tropical wave from the west coast of Africa that quickly moved across the eastern Atlantic where it rapidly intensified. Hurricane Irma reached category 5 status as it traveled through the Caribbean and maintained category 4 classification as it made landfall in Florida, affecting the entire state. Elevated wind and rain from Hurricane Irma passed through St. Augustine, a coastal city in northeastern Florida, on 10 September 2017 to early the morning of 11 September 2017. Winds reached a maximum speed of approximately 65 mph with 168 mm of precipitation [26]. Even though the eye of the hurricane did not pass directly through this area, this part of the state was still greatly affected by the storm with a surge reaching 0.9–1.5 m [26].

This study was conducted along Pellicer Creek, a major tributary of the Matanzas River within the Guana Tolomato Matanzas National Estuarine Research Reserve (GTM-NERR) in St. Augustine, Florida. Monitoring locations used in this study span the aquatic continuum between a fresh black water river and an estuary. The estuarine portion of the site is located between the Whitney Laboratory for Marine Biosciences and the Princess Place State Park, and the blackwater river, Pellicer Creek, is located within the Pellicer Creek Aquatic Preserve that flows through Faver-Dykes State Park and spans from Princess Place State Park and beyond the freshwater sites for this study. The brackish piece of Pellicer Creek is represented as middle reaches (Figure 1). The estuarine portion of the study site can reach salinities as high as 40 psu, whereas the farthest edge of the study site is primarily fresh water (0 psu, besides long drought periods with low flows) and is designated by freshwater 1 and freshwater 2 sites. Pellicer Creek is a major tributary of the Matanzas River. Bathymetry data collected in April 2016 by the GTM-NERR in Pellicer Creek's middle reaches, displayed an average depth of 1.7 m with a range from 1.3 to 2.1 m. The Matanzas River at the estuary location has an average depth of 1.5 m with a range from 0.8 to 1.8 m. Average tidal range at the middle reaches of Pellicer Creek was approximately 0.45 m with a range from 0.35 to 0.55 m, and the average tidal range of the estuary site at the Matanzas River was 0.55 m with a total range from 0.4 to 0.7. Since this data was only collected at one time point, there might be more variability based on seasonality and storm events [29]. This region is characterized by southern USA temperate climate with an average annual rainfall of 140 cm and a summer wet season between June and September. Pellicer Creek experiences semidiurnal tides with an average range of 0.6 m [16].

### 2.2. Short Duration High-Resolution Water Quality Monitoring

Three multi-parameter sondes (YSI EXO 2, Yellow Springs Instrument Company, Yellow Springs, Ohio, USA) were deployed across the study site. One was located in the estuary, another in the middle reaches between the estuary and the freshwater river, and the third was located are approximately 10 m inland from the estuarine location (freshwater 1; Figure 1). Each of the sondes were configured to record data at 15-min intervals. Sondes at the estuary and freshwater 1 sites, were calibrated to established standards [30], and were deployed before the storm on 7 September. The YSI EXO 2 sonde within the estuary was near the top of the water column on a floating dock (0.6 m), whereas the YSI EXO 2 sonde located at the freshwater 1 site was attached to the bottom of the riverbed due to lack of other attachment surfaces (1.1 m deep). At the middle reaches site a YSI 6600 sonde (owned by the GTM-NERR) is semi-permanently installed near the top of the water column (0.5 m) on a floating dock and continuously collects data that is posted publicly on the surface water monitoring program (SWMP) NERR site [31]. Additional meteorological data, including precipitation data, is collected and available on the same site. All three sondes had the ability to log temperature, conductivity, salinity, pH (although data at the estuary location is unavailable due to probe malfunction), and dissolved oxygen (DO). The sonde located in the middle reaches additionally measures relative depth, and the two YSI EXO 2 sondes (estuary and freshwater 1 sites) also logged total Chlorophyll-*a* (Chl-*a*), blue-green algae, turbidity, total dissolved solids, and fluorescent dissolved organic matter (fDOM).

Interference from high turbidity concentrations has been shown to cause issues with chlorophyll and fDOM measurement accuracy, but YSI EXO instruments have an error of only 0.5 µg L$^{-1}$ chlorophyll interference at 100 NTU as opposed to the older instruments that were 3 µg L$^{-1}$ at 100 NTU [32]. Therefore, it is assumed that interference was minimal (11 September 2017).

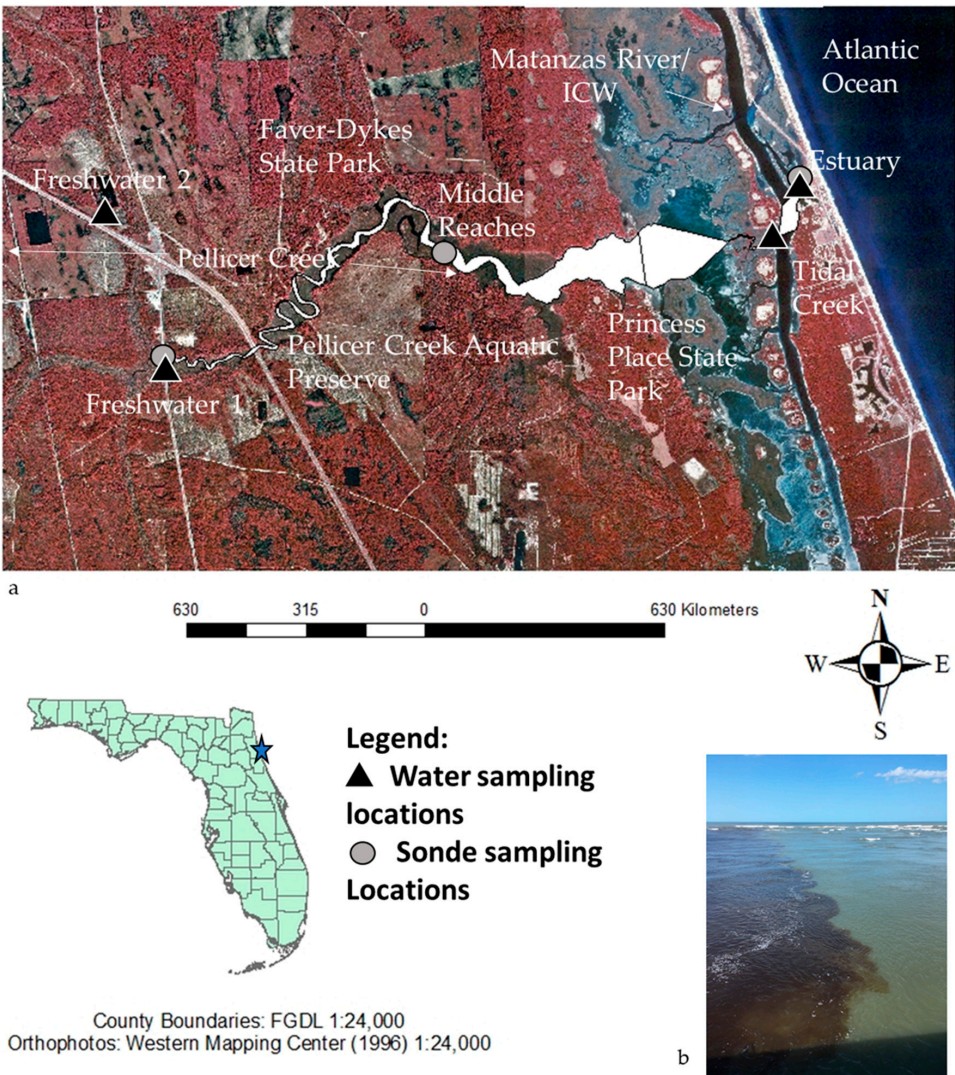

**Figure 1.** The three locations where YSI data sondes were placed along the study site located in St. Augustine, FL (grey circles). (**a**) Map was created using ArcGIS 10.5 software (ESRI). Coordinate System: GCS WGS 1984. The baselayer was created from orthophotos from the Western Mapping Center. Estuary location is location within the Intracoastal Estuary, middle reaches is in Pellicer Creek blackwater river that impacts from daily tidal cycles, and freshwater 1 is located near the headwaters of Pellicer Creek where salinity drops to 0 psu. Water samples for longer-term sampling were collected at the estuary, tidal creek, freshwater 1, and freshwater 2 sites (black triangles). The span of the study site from the estuary to freshwater 2 is approximately 15 km in length when following the waterway. The nearest inlet between the estuary and the Atlantic Ocean is approximately 4.5 km south. (**b**) The nearest inlet between the estuary and the Atlantic Ocean (bottom right) is approximately 4.5 km south. Precipitation during the hurricane resulted in export of highly turbid waters to the ocean.

Data download was conducted manually from the sondes at the estuary and freshwater 1 sites on 22 September, however the freshwater 1 sonde malfunctioned and only logged data from 7 to 12 September. The estuary sonde collected data from before the storm (7–10 September), during the storm (10 September, 8 p.m. E.T.–11 September, 1 a.m. E.T.), and after the storm until the sonde was removed

from the waterway (11–22 September). Surface water discharge data was downloaded for the middle reaches and is posted publicly on the United States Geological Survey's national water information system website (Figure S1; unpublished data) [33].

### 2.3. Long Duration Monitoring of Carbon, Phosphorus, and Metals

Over a period of 3 months, every 2–3 days initially post-storm and then weekly for the remainder of the study, water samples were collected at four sites across the study site, including the estuary, tidal creek, freshwater 1, and freshwater 2 sites (Figure 1). Water samples were not able to be collected near the middle reaches of the site due to park and road closings for several months post-hurricane. Grab samples were taken by hand in high-density polyethylene 1 L bottles pH and salinity were measured, then filtered with 0.45 μm filters, acidified, subsampled, and frozen until analysis. Samples were analyzed for DOC on a Shimadzu TOC-L Total Organic Carbon Analyzer (Colombia, MD, USA). Phosphorus was analyzed on a Hach DR 6000 spectrophotometer (Loveland, CO, USA) following EPA method 365.3. Subsamples from the estuary site and freshwater 1 were sent to the University of Florida Analytical Research Laboratories (Gainesville, FL) for total dissolved cadmium, copper, iron, molybdenum, lead, and zinc analyses on a Perkin Elmer Optima 5300 inductively coupled plasma spectrophotometer (Perkin Elmer Inc., Akron, OH, USA). It should also be noted that at the beginning of October, during the duration of monitoring, a nor'easter event (storm with north-easterly wind occurring on the east coast of North America) passed through the study site.

### 2.4. Statistical Analysis

Graphical comparisons of salinity with meteorological data, salinity values across locations, turbidity vs. Chl-*a* at the estuary location, and fDOM at the estuary and freshwater 1 location were conducted to initially visualize data trends. Some data did not meet the assumptions of linearity or normality to utilize a linear regression, therefore Spearman correlations were employed to evaluate these relationships on a subset of well-correlated parameters from before, during, and after the hurricane (Table 1). Parameters used in this analysis included salinity vs. DO for estuary and middle reaches sites, fDOM vs. salinity at the estuary site, and salinity vs. pH at the middle reaches site. The freshwater 1 sonde also malfunctioned the day after the hurricane and therefore did not record enough observations to make statistically valid comparisons of characteristics from before and during the storm to after the storm had passed.

**Table 1.** Correlations compared before, during, and after Hurricane Irma as well as overall for the chosen parameters. *p* values in parentheses.

| Location | Parameters | Before Irma | During Irma | After Irma |
|:---:|:---:|:---:|:---:|:---:|
| **Estuary** | DO vs. salinity | 0.323 (<0.01) | 0.980 (<0.01) | 0.941 (<0.01) |
| **Estuary** | fDOM vs. salinity | −0.845 (<0.01) | −0.984 (<0.01) | −0.955 (<0.01) |
| **Mid. Reaches** | DO vs salinity | 0.875 (<0.01) | 0.833 (<0.01) | −0.260 (<0.01) |
| **Mid. Reaches** | pH vs. salinity | 0.988 (<0.01) | 0.952 (<0.01) | 0.869 (<0.01) |

Principal component analysis (PCA; '*factoextra*' R-library) was used to compare salinity, DO, temperature, and turbidity collected at all three sites by data sondes. PCA was performed to determine how closely the parameters were related at these three sonde locations.

Data for salinity, pH, total dissolved phosphorus, total dissolved iron, and dissolved organic carbon collected and measured from water samples were normalized for T = 0 and graphed to display changes from the original state over time in Microsoft Excel.

Additionally, a Spearman's rank correlation was performed between fDOM and DOC data in R package "ggpubr" collected on 22 January 2018 at 20 locations at both high and low tide across the study site in order to assess how fDOM relative fluorescence relates to DOC within this system. This relationship can be used to link fDOM to DOC (and indirectly DOM) concentrations during sonde

deployments since water sampling was not possible at the same interval as the sonde measurements. Even though fDOM is only a small fraction of total DOM, fDOM is used as a proxy for the changes in DOM concentrations within the study site since it was not possible to measure total DOM continuously on the same timescale.

Statistical operations were performed with R© (Ver. 3.1.2, R Foundation for Statistical Computing, Vienna Austria), and unless otherwise stated all statistical operations were performed using the base R library. The critical level of significance was set at $\alpha = 0.05$.

## 3. Results

### 3.1. Short-Term Water Quality Monitoring of Hurricane Response (St. Augustine)

Under normal ambient pre-hurricane conditions (measured 7–10 September), salinity values in the middle reaches fluctuated with daily tidal cycles. These cycles have a wide seasonal range, 10–30 psu during high tide and 0–20 psu at low tide. However, during Hurricane Irma, salinity levels peaked as wind speeds reached approximately 31 m s$^{-1}$ (65 mph; Figure 2), then decreased rapidly as total precipitation reached a peak of approximately 24 mm in 1 h. For approximately 5 days after the storm, salinity levels in the middle reaches remained close to 0 psu and then slowly began to return to pre-storm levels (approximately 0–20 psu range) by day 10 post-storm. At the estuary, salinity levels also declined by 10–20 psu after Hurricane Irma, but daily fluctuations in salinity show that tidal cycling still affected salinity in this area (Figure 3). Lower salinity levels than pre-hurricane levels (predominately < 30 psu) were also still observed at this location for several weeks after the hurricane. The freshwater 1 sonde (15 km inland) salinity remained constant at 0.1 psu except at the storm's peak where concentrations rose to 2 psu, indicating some saline intrusion into freshwater reaches of the creek.

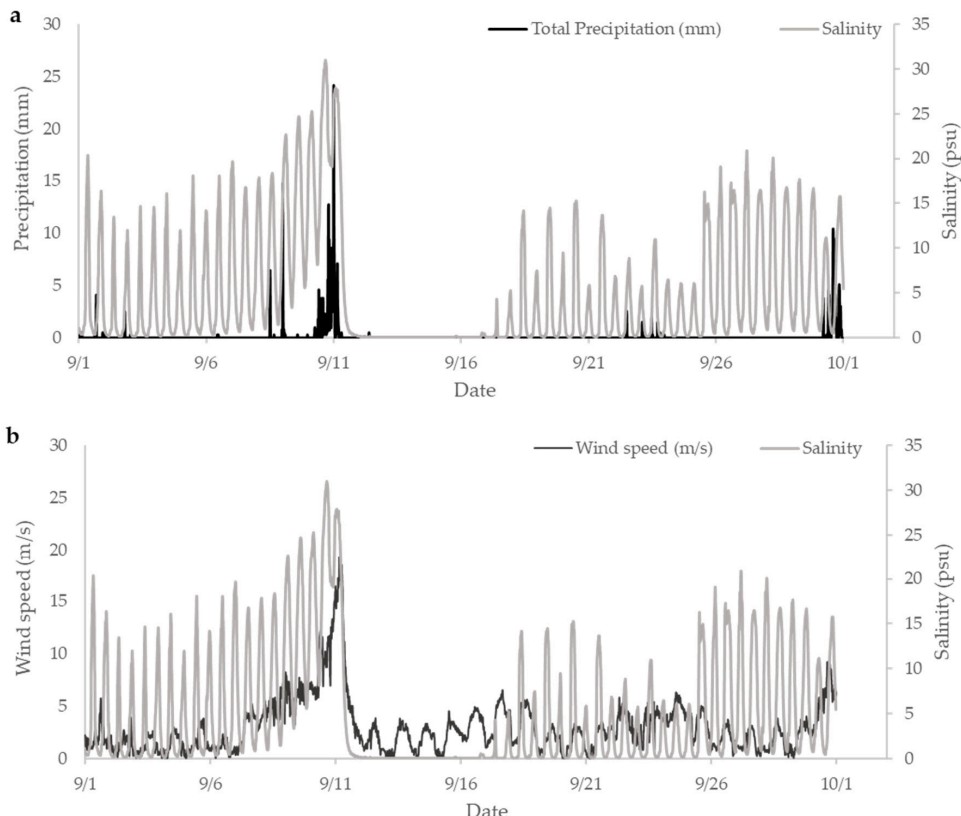

**Figure 2.** Wind speed (**a**) and precipitation (**b**) measurements taken during the month of September at the middle reaches site and the effects on salinity concentrations within the study site. The spike in precipitation and wind speed indicates the time of impact from Hurricane Irma on the study site.

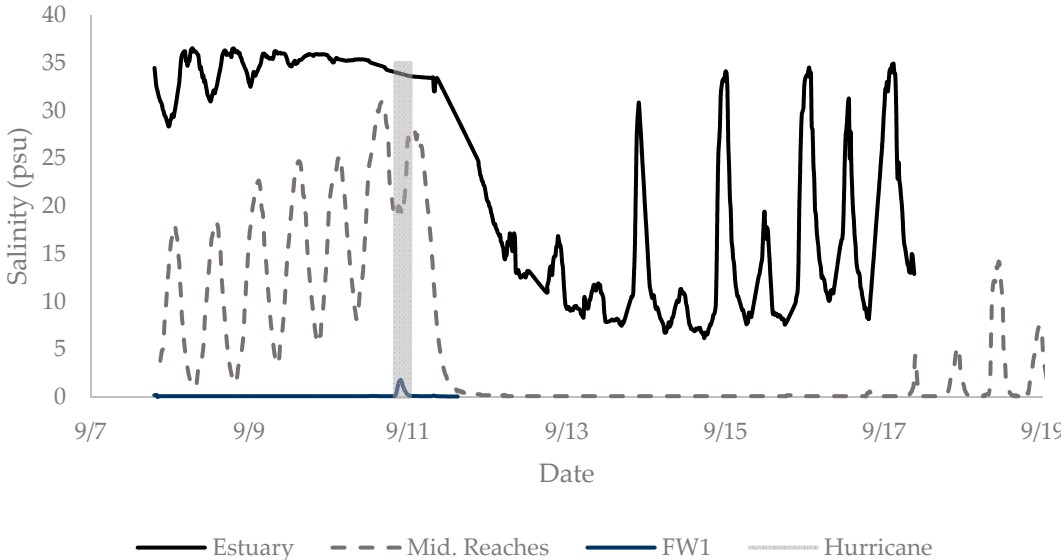

**Figure 3.** Salinity graph showing fluctuations at the three sonde locations for before, during, and after Hurricane Irma. Salinity values within the estuary (black line), middle reaches (dashed grey line), and freshwater 1 (solid grey line) 2 days before Irma, during the storm (grey bar), and 1 week after the hurricane had passed.

Turbidity levels began increasing during the hurricane, peaked immediately after it passed, and remained elevated for approximately 2 days after the hurricane made landfall in the St. Augustine area (Figure 4). Chl-*a* levels decreased substantially just prior to the storm and steadily increased after the storm once turbidity levels began to drop. However, turbidity and Chl-*a* were not significantly correlated ($r_s = -0.07$, $p > 0.05$) suggesting that other factors such as pulses of nutrients during and/or after the storm enhanced aquatic primary production.

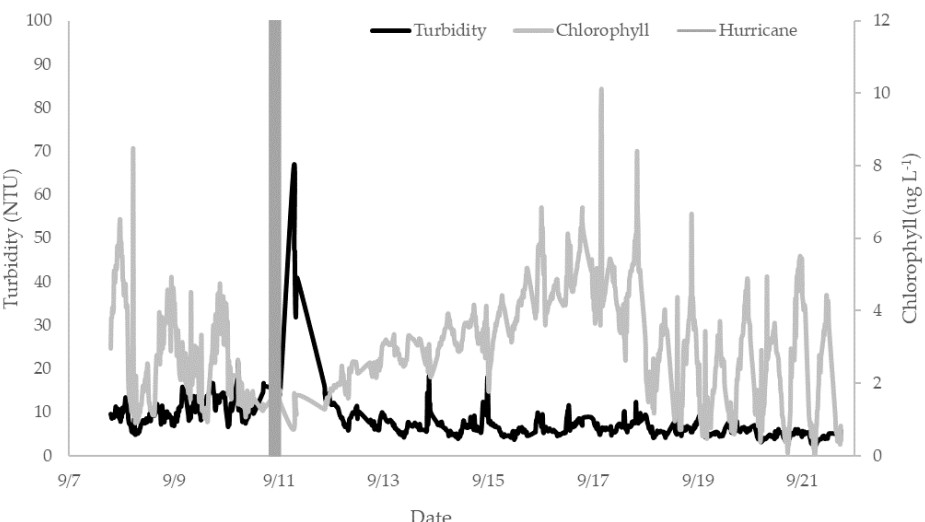

**Figure 4.** Turbidity and Chlorophyll-*a* measurements taken at the estuary sonde location during the storm. Chlorophyll data collected between the apex of the storm on 10 September 2017 and 12 September 2017 might be inaccurate due to potential interference from high turbidity levels.

Fluorescent DOM values measured at freshwater 1 increased marginally during the hurricane until the sonde malfunctioned after 5 days (Figure 5). In the estuary, fDOM values were affected widely by tidal cycles and were much lower before and during the storm but increased greatly after the hurricane from 100 quinine sulfate units (QSU) to almost 200 QSU. During high tide, fDOM values

were much lower than at low tide when high concentrations of fDOM were being carried into the area by freshwater pulses from upstream.

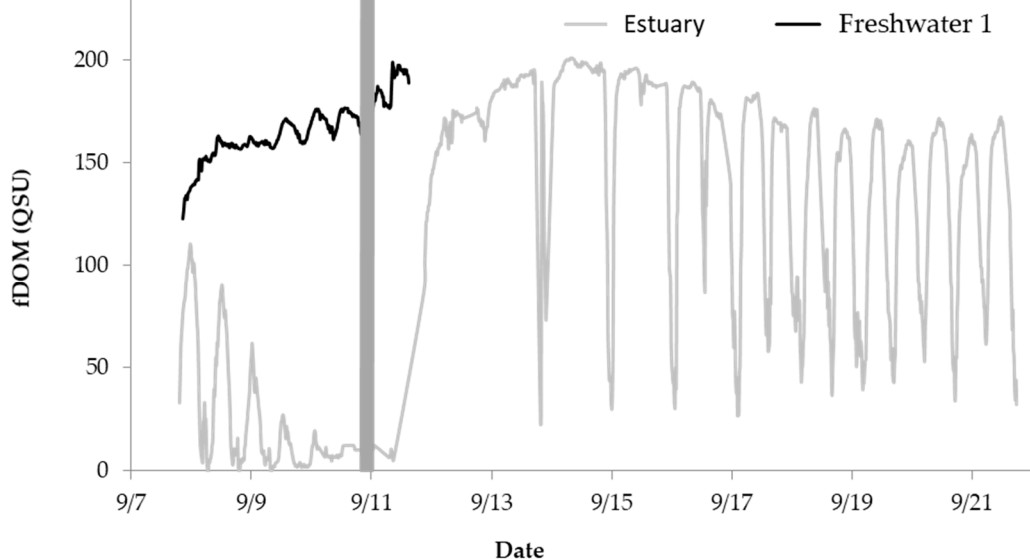

**Figure 5.** fDOM concentrations at the estuary and freshwater 1 sonde locations. The sonde located at the freshwater 1 site malfunctioned after the storm, so only 4 days of data were successfully recorded.

Dissolved oxygen ranges varied widely across the three sites. DO concentrations were highest at all the estuary and middle reaches sites before the storm, whereas concentrations increased during the storm until the sonde malfunctioned at the freshwater 1 site (Figure 6). DO within the estuary ranged from 70% to 100% pre-storm, then displayed a wider range of concentrations post-storm yielding concentrations as low as 30% and as high as 100%. DO ranged from 40% to 90% pre-storm in the middle reaches of the site but decreased steadily post-storm to concentrations remaining below 50% for the duration of the study. Temporary hypoxia from DO concentrations as low as 6.9% were seen 1 week after hurricane Irma. DO concentrations at the freshwater 1 site ranged from 30% to 60% in the few days before instrument malfunction and peaked at 61% as the storm passed through the site. These measurements appear to be tidally influenced and therefore additionally vary based on the daily tidal cycle.

Salinity and DO were highly correlated at various points pre- and post-storm at estuary and middle reaches locations (Figure 7). Pre-storm middle reaches DO values were above 40% saturation, and salinity had a wide variability, most likely due to location within the waterway ($r_s = 0.875$, $p < 0.001$; Table 1). A strong positive relationship was noted between % DO and salinity ($r_s = 0.833$, $p < 0.001$). After the storm, DO concentrations decreased from 40%–90% saturation range to 5%–60% saturation range and salinity decreased from a range of 0–20 psu to a stable 0 psu ($r_s = -0.260$, $p < 0.001$). In the estuary, DO and salinity values were concentrated in the high ranges for both parameters before the storm, exhibiting a weak positive correlation ($r_s = 0.323$, $p < 0.001$). Within the estuary, DO and salinity were positively correlated during the hurricane ($r_s = 0.980$, $p < 0.001$), and remained positively correlated after the storm ($r_s = 0.941$, $p < 0.001$). Parameters at freshwater 1 were not compared due to a sonde malfunction post-storm.

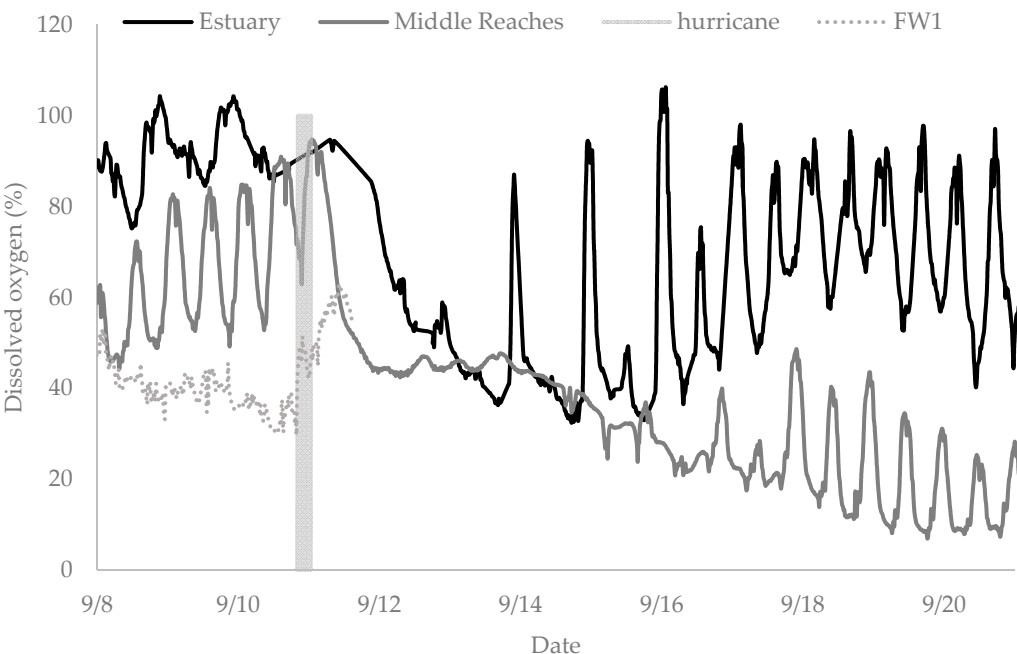

**Figure 6.** Measured fluctuations in dissolved oxygen across the study site. The estuary and middle reaches sites data spans from 3 days before to approximately 10 days after Hurricane Irma. DO data from the freshwater 1 site is shown for the first few days of deployment before the sonde malfunctioned post-storm.

Salinity and pH had distinctive trends and were compared for pre-storm, during storm, and post-storm correlation (Figure 7d). Salinity and pH were compared in middle reaches only (due to probe malfunction at the estuary) and positively correlated before and during the hurricane (before $r_s = 0.988$, $p < 0.001$; During $r_s = 0.952$, $p < 0.001$). Post-storm, a decrease in pH and salinity reduced the range of values and decreased the correlation coefficient to $r_s = 0.869$, $p < 0.001$. Salinity and fDOM were compared in the estuary and displayed a highly negative correlation ($r_s = -0.845$, $p < 0.001$). Significant correlations of both parameters during and after Hurricane Irma exhibited a negative trend of $r_s = -0.984$, $p < 0.001$ and $r_s = -0.955$, $p < 0.001$ during and after the storm.

Principal component analysis of study site sonde data indicated varying relationships between salinity, DO, temperature, and turbidity at estuary, middle reaches, and freshwater 1 locations (Figure 8). PCA yielded four components and the plot generated illustrates that most of the variability is within the first two principal components that explain 61.7% and 27.6% of the variability. Other principal components each explain less than 10% of the overall variability. Overlap of the middle reaches and estuary components indicate some commonality among sites, whereas the freshwater 1 site appears to be unique and does not share commonalities with the other two locations.

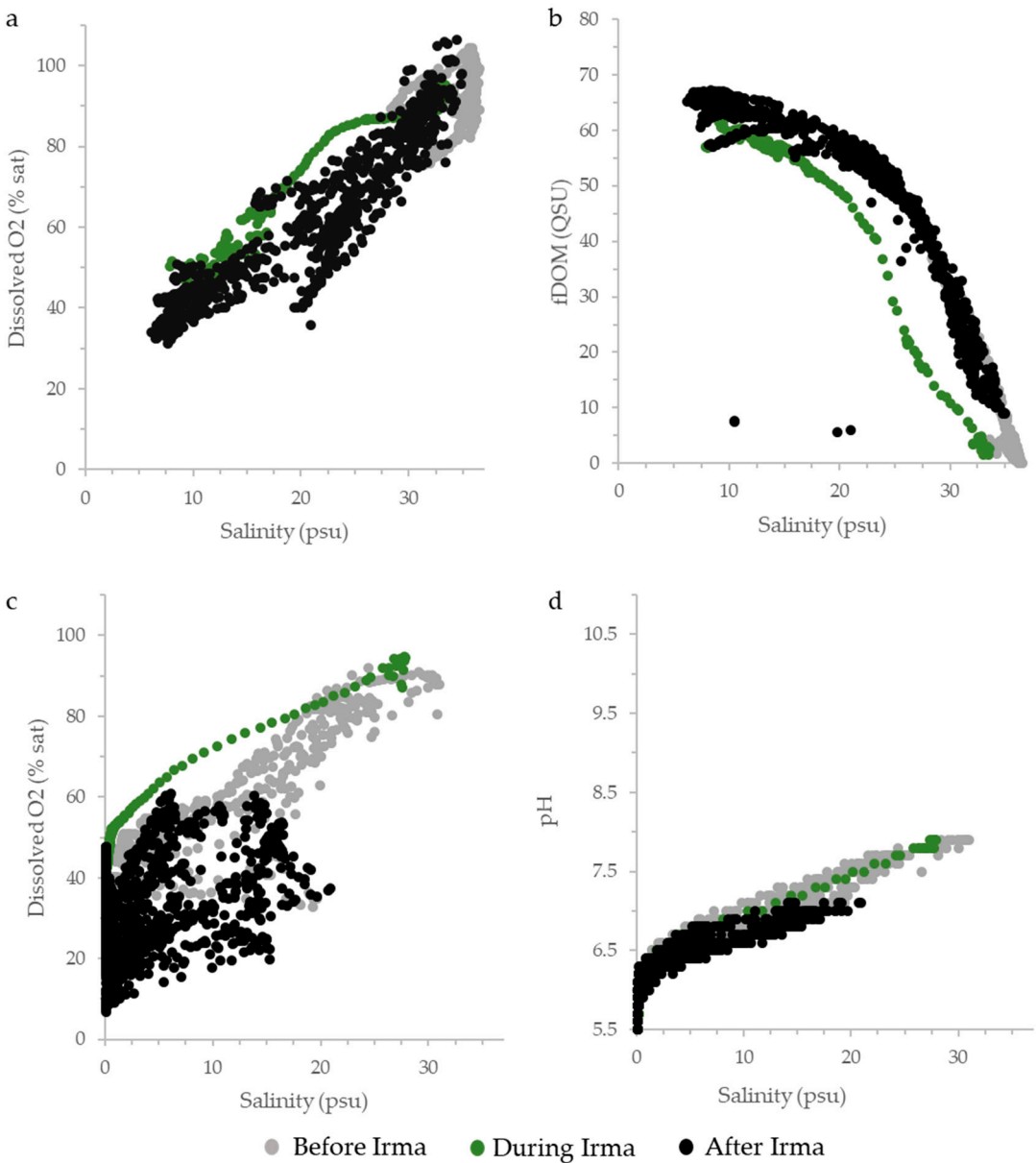

**Figure 7.** Correlations from before (grey), during (green), and after (black Hurricane Irma compared for a series of parameters: (**a**) dissolved oxygen and salinity within the estuary, (**b**) fDOM and salinity at the estuary, (**c**) dissolved oxygen and salinity in the middle reaches, (**d**) pH and salinity in the middle reaches. This data was collected by YSI data sondes from 7 September to 22 September 2017. * Hurricane Irma affected the study area between 10 and 11 September 2017.

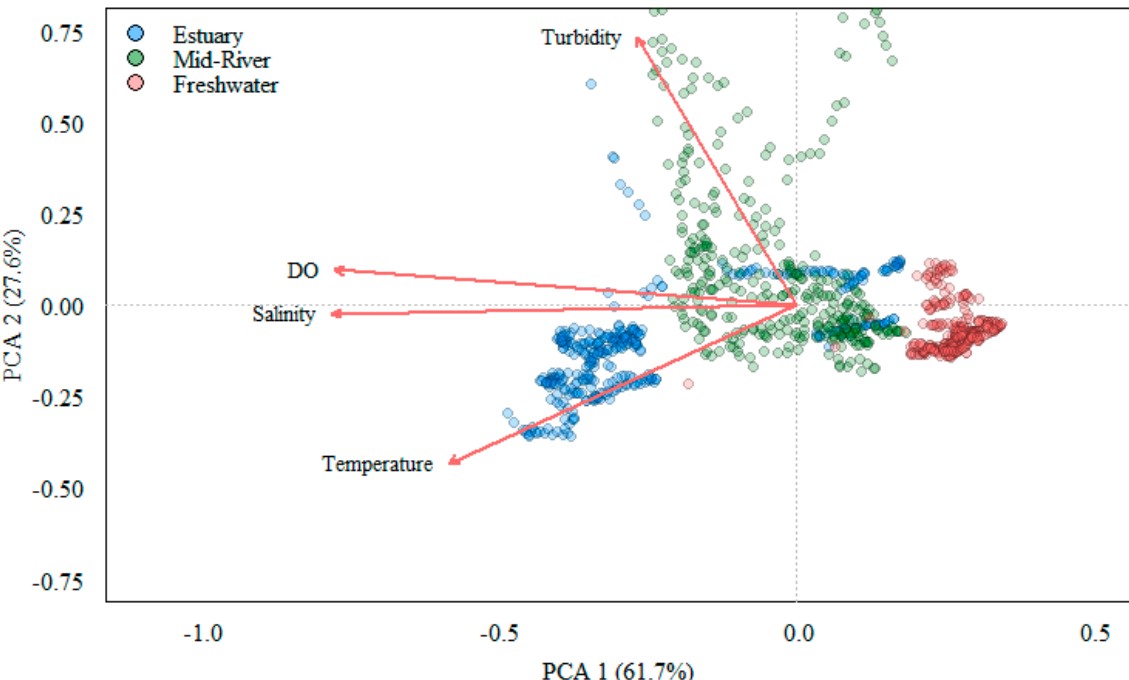

**Figure 8.** PCA biplot depicting the amount of correlation amongst water quality parameters salinity, DO, temperature, and turbidity during Hurricane Irma at the estuary (**blue**), freshwater 1 (**pink**), and middle reaches (**green**) sonde sites.

### 3.2. Long-Term (3 Month) Monitoring of Dissolved C, P, and Metals

Normalized salinity and pH values collected from water samples across the study site showed large differences over time (Figure 9a,b). Salinity from the estuary and tidal creek sites (only locations where sal > 0 psu), decreased initially and fluctuated across time from pre-storm conditions until returning to approximately pre-storm conditions 60 days after Hurricane Irma's passage. pH fluctuated initially in the estuary, tidal creek, and freshwater 1 sites, but returned to pre-storm conditions after approximately 40 days. The freshwater 2 site showed a different trend and increased over time across the entire length of the study.

Total dissolved phosphorus concentrations were measured at the four sites every 2–3 days in the week post-storm and weekly for the next approximately 3 months, but only data from the freshwater sites showed concentrations above the 0.02 mg $L^{-1}$ detection limit and are shown in Figure 9 (Figure 9c; Table 2). Measurable values were collected at the freshwater 1 site on 7 September (pre-hurricane), 16 September, 18 September, 21 September, 28 September, and 12 October (just over 1 month post-Irma). Concentrations ranged between 0.15 and 0.3 mg $L^{-1}$. Freshwater 2 site did not display any measurable concentrations until November (pre-storm sample was not taken at this site) but ranged from 0.2 to 0.4 mg $L^{-1}$ for three measurements between 2 November and 16 November.

Total cadmium, copper, iron, molybdenum, lead, and zinc were measured at the estuary and freshwater 1 sites due to funding limitations during the 3-month sampling period. Iron was the only metal yielding any measurable values above the minimum detection limit at the freshwater 1 site and showed variability in concentrations over time ranging from approximately 0.5 to 1.6 mg $L^{-1}$ (Table S1; Figure 9d). Iron concentrations returned to pre-hurricane conditions approximately 1 week post-hurricane, then decreased again to below the minimum detection limit after the nor'easter event for approximately 1 week before returning to pre-hurricane concentrations again (Table 2). Iron concentrations decreased again at the end of the study period even though there were not any large storm events during this period.

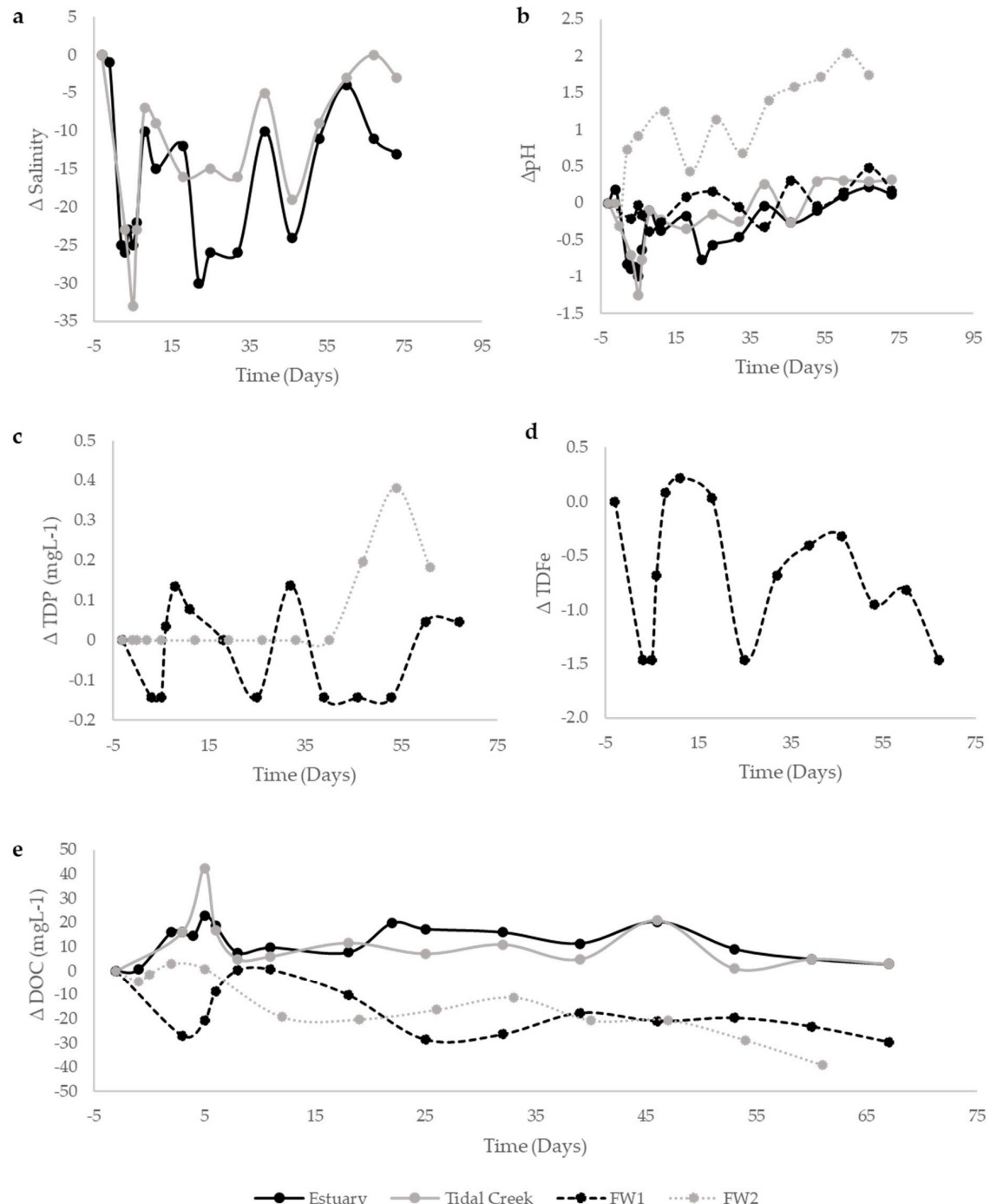

**Figure 9.** Measured concentrations for changes in (**a**) salinity, (**b**) pH, (**c**) total dissolved phosphorus, (**d**) total dissolved iron, and (**e**) dissolved organic carbon at estuary, tidal creek, freshwater 1, and freshwater 2 locations from initial concentrations (collected on 7 September 2017). All values were normalized to time point 0 values to display the differences from initial conditions over time. Salinity was 0 psu across all measured time points at freshwater 1 and 2 sites, and total dissolved phosphorus and iron concentrations were below the minimum detection limit at the estuary and tidal creek locations. Hurricane Irma passed through the study site 3 on day 3 of the study.

**Table 2.** Total dissolved phosphorus (TDP) and total dissolved iron (TDFe) concentrations from collected water samples that yielded values above the minimum limit of detection. Although other time points yielded dissolved organic carbon (DOC) values above the minimum detection limit, measurements from the specific dates where TDP and TDFe was measurable was included as a reference.

| Site | Date Collected | TDP (mg L$^{-1}$) | TDFe (mg L$^{-1}$) | DOC (mg L$^{-1}$) |
|---|---|---|---|---|
| Freshwater 1 | 7 September 2017 | 0.153 | 1.49 | 59.73 |
| | 16 September 2017 | 0.189 | 0.81 | 51.1 |
| | 18 September 2017 | 0.289 | 1.57 | 60.12 |
| | 21 September 2017 | 0.231 | 1.71 | 60.32 |
| | 28 September 2017 | 0.153 | 1.52 | 49.65 |
| | 12 October 2017 | 0.290 | 0.81 | 33.3 |
| | 19 October 2017 | 0.02 [a] | 1.08 | 42.18 |
| | 26 October 2017 | 0.02 [a] | 1.17 | 38.76 |
| | 2 November 2017 | 0.02 [a] | 0.54 | 40.16 |
| | 9 November 2017 | 0.002 [a] | 0.67 | 36.5 |
| Freshwater 2 | 2 November 2017 | 0.206 | - | 44.34 |
| | 9 November 2017 | 0.392 | - | 36.02 |
| | 16 November 2017 | 0.194 | - | 25.74 |

[a] Value reported as less than the minimum detection limit (MDL), therefore data were set to the MDL for analysis purposes.

Carbon concentrations fluctuated over the nearly 3-month period of monitoring (Figure 9e). Pre-storm concentrations were nearly 60 mg L$^{-1}$ at the freshwater 1 site (freshwater 2 not measured this day) and approximately 1 mg L$^{-1}$ in the estuary and tidal creek sites (Figure S2). By 13 September 2017 (2 days post-hurricane) the freshwater 1 site had a value of nearly half its pre-storm conditions (32.7 mg L$^{-1}$), although the freshwater 2 site DOC was nearly 65 mg L$^{-1}$. The estuary and tidal creek DOC concentrations increased by at least 15 times and were between 15 and 20 mg L$^{-1}$ DOC at this time period. The estuary and tidal creek sites reached the maximum measured DOC concentrations by 16 September 2017 (approximately 5 days post-hurricane) at 18 and 43 mg L$^{-1}$. DOC concentrations rebounded back up to 60–70 mg L$^{-1}$ in the freshwater sites by 22 September 2017 (approximately 11 days post-storm) as the estuary and tidal creek sites DOC decreased below 10 mg L$^{-1}$. The estuary and tidal creek sites did not fully return to normal concentrations before a nor'easter affected the study site the first week in October, causing enough precipitation to increase water discharge levels to $8 \times 10^6$ m$^3$ day$^{-1}$, even though only $6 \times 10^6$ m$^3$ day$^{-1}$ was actually measured during Hurricane Irma (Figure S1). The nor'easter also caused freshwater 1 site concentrations to decrease to 30 mg L$^{-1}$ DOC and the freshwater 2 site to 45 mg L$^{-1}$ DOC. The nor'easter event likely prolonged the length of time required for the system to return to pre-hurricane conditions and the last measurements taken in November yielded concentrations around 4 mg L$^{-1}$ at the estuary and tidal creek sites and concentrations 25–30 mg L$^{-1}$ at the two freshwater sites.

As expected, fDOM and DOC concentration are strongly correlated with a moderate ρ value observed during this study (Figure 10; ρ = 0.69, *p* = $7 \times 10^{-7}$). During pre-hurricane, modeled DOC concentrations range from approximately 0 to 60 mg L$^{-1}$ and post hurricane concentrations were an order of magnitude higher and remained high post-Hurricane in the estuary. Meanwhile modeled DOC concentrations and fDOM values continued to increase post hurricane at the freshwater 1 site (Figure 5).

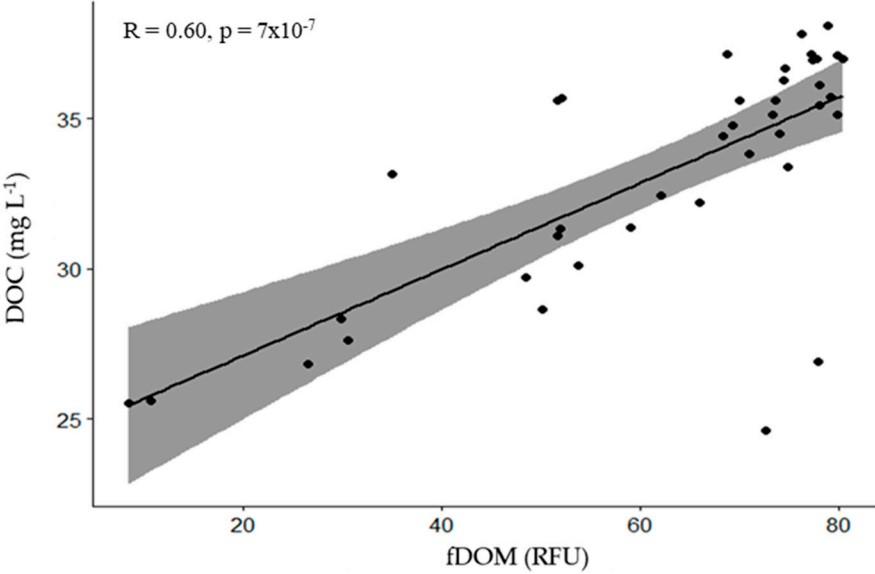

**Figure 10.** A Spearman's rank correlation was run fDOM and DOC concentrations collected on 22 January at 20 points across the study site at low and high tide. Spearman correlation ρ was measured at 0.69 and indicates a moderate relationship between the two variables.

## 4. Discussion

### 4.1. Biogeochemical Responses to Hurricane Irma

Salinity and changes in DO concentrations were the primary water quality factors affected during and immediately after Hurricane Irma which are presumably driven by storm surge, freshwater flushing, wind driven aeration, and biological oxygen demand. Hurricanes can have profound effects on ecosystem characteristics and structure. In the late 1990s in Pamlico Sound, North Carolina (USA) a hurricane caused the estuarine residence time to decrease from 1 year to 2 months and salinity to drop by 70%, driven by extremely high discharge volumes from the rivers and stream entering the sound [34,35]. Although instrumentation was not available at the time of this storm, increased discharge by an order of magnitude relative to baseflow conditions resulted in decreased salinity by as much as 14 times. This same hurricane dynamic was evident in the Pellicer Creek/Matanzas estuary post Hurricane Irma whereby large amounts of freshwater discharge resulted in increased DOM abundance and decreased DO concentrations.

Additionally, flushing causes a pulse of DOM during storms that represents rapid mobilization of organic material from the terrestrial landscape and adjoining aquatic habitats, and in some cases these rapid events may shunt DOM to downstream reaches of the ecosystem with little alteration or change in chemical structure, a phenomenon described as the pulse-shunt concept [36]. However, abrupt DOM loading can also result in depressed DO concentrations from increasing biological oxygen demand, which has been observed during prior large storms including Hurricanes Hugo and Charley [37–39].

Prolonged discharge levels effecting nutrient transport and salinity can also alter phosphorus concentrations within the water column. One study in the Neuse River Estuary in North Carolina observed that during tropical storms with high precipitation, total phosphorus loads increased by 25.7% [14]. P concentrations remained below the limit of detection at the estuarine site, and unlike the Neuse River study increases in P loads were not observed downstream in our study presumably due to uptake, dilution, binding, or flocculation from increases in salinity during transport downstream [40]. Reducing species of OM through redox reactions can also cause binding of phosphate and ammonium with DOC that could have been caused by decreasing oxygen levels [41]. Reoxygenation of the water column post-hurricane, mass transport of nutrients due to increases in rainfall-runoff, and delayed mortality and decomposition of organic matter could have caused a delayed release of nutrients into the

water column, resulting in mass algal uptake of these released nutrients and a delayed algal response to the hurricane, causing a steep rise in Chl-*a* concentration 1 week post-storm [23,42]. During this study, total dissolved phosphorus (TDP) concentrations were below the detection limit at the estuary site despite relatively higher concentrations observed at upstream freshwater sites (Table 2). This gradient of P is the result of an integrated function between availability, demand, recycling, and transport. In stream ecology, this function was coined nutrient spiraling where during transport downstream nutrients are taken-up, assimilated, and recycled [43,44]. Changes in factors that contribute to nutrient spiraling (i.e., biotic uptake, export, and velocity) can influence an ecosystem's ability to process and retain nutrients resulting in pulses of nutrient from the freshwater reaches of the system to the downstream estuary. This potentially flushes nutrients, such as phosphorus, and temporarily high algal loads out of the estuary relatively quickly (Figure S1).

Moreover, as nutrients rapidly flow downstream forcing conditions to shift to a reduced redox state, iron availability and cycling can be significantly altered. In oxidizing (aerobic) environments, iron can be tightly bound to DOC. However, under anaerobic conditions, iron becomes reduced and DOC can flux into the water column [45]. Pyrite formation is also possible under these conditions where iron could be readily scavenged from the water column as it is being reduced [46]. In this study anoxic conditions persisted post-hurricane for 2 weeks it is likely that anoxic conditions occurred in the middle (and likely upper-reaches) of the study area as DO decreased to levels as low at 5%. Iron concentrations mostly measuring above 1.5 mg L$^{-1}$ within the water column in the freshwater 1 site in the month post-hurricane may be indicative of this type of reaction. Meanwhile, iron concentrations were below the minimum detection limit (0.05 mg L$^{-1}$; Table S1) in the estuarine environment, indicating the possibility of rebinding to DOC as it was oxidized by new aerobic waters entering the system through tidal flushing, burial due to pyrite formation or, alternatively, due to flocculation of both species with OM as a result of changes in salinity [40].

Carbon dynamics during and after a hurricane can be complex as normally passive channels of transfer from land to ocean become very active [2,47,48]. Storm events cause both a peak in discharge volume from tributaries but also a peak in DOC (and DOM) loads with levels remaining elevated for several weeks post-disturbance as terrestrial and wetland systems slowly drain into larger water conveyance systems [49]. Increased DOC downstream post-storm from flushing upstream was caused by large increases in sheet flow and soil infiltration that funneled a large portion of carbon into the estuary. These dynamics observed in the Pellicer Creek/Matanzas Estuary during Hurricane Irma further supports the importance of the pulse-shunt concept within this system [36,50,51]. The high freshwater post-storm runoff from upstream in response to prolonged and abundant rainfall pre- and during hurricane landfall (Figure S1) carrying with it a significant DOC pulse across the study area persisting for several days (months) post hurricane. The discharge from middle reaches post-storm continued for approximately 50 days until returning to pre-storm discharge (Figure S1; unpublished data, United State Geologic Survey) and contributed to a prolonged load of DOC to the estuary. However, the nor'easter event at the beginning of October that caused a second peak in discharge prolonged this increased loading. The discharge measurements collected for Hurricane Irma and the following nor'easter were the largest recorded at this site in the past 20 years (Figure S1a). Increases in river DOM concentrations due to storms both small and large are well-documented, and in some cases can contribute to over 90% of the annual flux of DOC from a system [52]. In the case of normal weather patterns, DOC levels typically return to baseline conditions shortly after discharge recedes [53]. However, unlike our observations after Hurricane Irma where after 3 months DOC concentrations at freshwater sites had not yet fully returned to pre-storm levels. Dilution and large flushing events might have transported large portions of DOC from the upper reaches to downstream locations, so it is possible that the system takes longer than 3 months to rebuild to previous concentrations [14].

Relationships between salinity, fDOM, and pH displayed high but variable levels of correlation across time (Figure 7a–c). Salinity and pH in the middle reaches showed a decreased level of correlation post-storm due to higher variability in DOM and reduced buffering by seawater, both well-known

drivers of pH [54,55]. The relationships of fDOM, salinity, and pH observed in the downstream estuary and marine ecosystems are, in part, influenced by the upstream system. The upstream ecosystem, a blackwater creek, has a high level of connectivity with its terrestrial components and associated wetlands, which contribute to its characteristics (i.e., high fDOM load and turbidity). This connectivity facilitates the free exchange of OM between the creek and wetlands and smaller (tributary) creeks within the watershed. Normally, the OM is delivered along the river continuum either slowly as the creek flows or in short pulses in response to rain events [36,48]. Under normal conditions, given the creek's length and hydrodynamic, the residence time of the creek allows for this OM (and associated nutrients) to be utilized as it moves along the river continuum. However large scale disturbances such as prolonged rain events and intense tropical activity effectively elongate the continuum and temporarily extend the freshwater portion of the river continuum into the estuary and marine portion, thereby influencing local biogeochemical cycles (OM degradation, nutrient availability, etc.), biotic interaction (i.e., oyster health, fish kills, etc.), and ecosystem condition (i.e., hypoxia, algal blooms, etc.) resulting in an altered stable or otherwise state. Therefore, largescale disturbances such as Hurricane Irma ultimately punctuate how the system functions in the context of the river continuum, ecosystem resilience to disturbance, and ecosystems resistance to change.

### 4.2. Implications on Ecosystem Resilience

Hurricane Irma's impact on several key water quality characteristics did not only affect biogeochemical cycling but may have similarly affected organism survival and health. Depressed DO levels post-Hurricane have led to prolonged anoxic and hypoxic conditions leading to widespread fish kills [37,56]. Oysters and various other aquatic organisms that can be sensitive to DO concentrations live across a large portion of the study site and are adapted to some salinity and DO fluctuations from daily tidal cycles, however even short-term hypoxia can have devastating effects on estuarine organisms [57]. Further, the harmful effects of hypoxia are compounded by other disturbances such as acidification [58], which also occurred during/after Hurricane Irma. In the months following Hurricane Irma, changes in oyster populations in the mid and lower reaches of Pellicer Creek estuary were observed and have been attributed to the extensive freshwater inputs and low oxygen conditions following the disturbance (Osborne, personal communication).

The observed spatial and temporal correlations of salinity and DO during this study are important with respect to the ecosystem's resistance and resilience to disturbance (Figure 7a,c, Table 1). A resilient system will return back to where it started after a disturbance relatively quickly whereas the end point of a less resilient system post disturbance would transition to an alternate "stable state" (i.e., not return to pre-storm conditions). This alternate stable state could be driven by landscape scale physical changes to the system (i.e., changes freshwater-marine exchange, changes to river width/drainage area, etc.) or biogeochemical alterations (i.e., changes in DOM inputs due to upstream impacts, changes in loading, etc.). The DO–salinity relationship observed here could serve as an indicator of ecosystem resilience with respect to large-scale disturbances as observed with Hurricane Irma. It is expected in a resilient, more flexible ecosystem that the shape of the DO–salinity curve post-disturbance would return to pre-disturbance conditions (hysteretic response), the pre-storm stable state. This hysteretic response and return to stable state is often linked back to ecosystem health, ecosystem resistance, and resilience to disturbance [59]. If a system does not experience this return, it may be less resilient and resistant and more likely remain in an altered stable state. When comparing the middle reaches and estuary locations (Figure 7), the lower end of the river continuum, it appears that the estuary hysterectically returns to its prior stable state, whereas middle reaches return to an altered condition. Given the volume of freshwater, prolonged period of low-DO concentrations, physical system alteration by the hurricane, and potential impacts due to eutrophication and sea-level rise, it is possible that the brackish middle reaches of the river continuum in this system is in a new stable (possibly temporarily degraded) state. Data collected from water samples further supports this statement, showing most measurable parameters, including DOC and pH, return to pre-hurricane conditions by the end of the study period

in the estuary and tidal creek sites (Figure 9). However, conditions in the freshwater sites showed differences in iron, pH (differences seen only in freshwater 2 site), DOC at the end of the study period that varied from pre-hurricane conditions. Since the freshwater 2 site is located near the headwaters of a forested blackwater river, pH is usually lower in this location than at the other sites. However, the increase in pH during the study may indicate a large flushing of organic matter from the freshwater 2 site that usually keeps this pH acidic. This indicates a difference in system resiliency across the site, freshwater sites with little tidal influence and daily fluctuation are taking much longer to return to the pre-storm conditions than the estuarine sites that are prone to large daily fluctuations.

Possibly, resiliency across the study site might also be related to the vegetation transition from salt-tolerant to freshwater species. The study site is surrounded by living shorelines that transition from mangrove habitat within the estuarine environment to saltmarsh and *Juncus romerianus* dominated systems and finally freshwater species (*Cladium jamaicense*, *Taxodium distichum*, etc.) in its upper reaches. Living shorelines, but mainly saltmarsh and mangrove habitats, have been shown to assist with ecosystem resilience during previous hurricanes and could help protect this ecosystem as well as the other NERR sites from large ecosystem shifts [60,61]. However, the impact of increased storm frequency on these ecosystems is still unknown and could add to future shifts in nutrient cycling and water quality, which requires more long-term monitoring with detailed measurements immediately before, during, and after the storm to determine the outlook for these ecosystems in the future.

## 5. Conclusions

In conclusion, water chemistry in the form of salinity, dissolved oxygen, DOM, and other constituents were temporarily disturbed across the aquatic continuum during Hurricane Irma. All measured parameters across locations affected one another as rapid changes occurred during the storm and some persisted for multiple weeks post-storm. Although not explored in this study, there are potential longer-term ecological repercussions from these rapid water quality changes from hurricane events that can affect ecosystem health and resilience, and those consequences need to be further examined to determine the potential implications of increased tropical storm intensity or frequency that may occur in the future.

**Supplementary Materials:** The following are available online at http://www.mdpi.com/2077-1312/8/6/412/s1, Figure S1: Pellicer Creek tidally filtered discharge, Table S1: Minimum detection limits, Figure S2: Concentrations of dissolved organic matter at four locations.

**Author Contributions:** Conceptualization, T.S., N.W., P.J. and T.Z.O.; Formal analysis, T.S., N.W. and P.J.; Investigation, T.S., P.J. and T.O.; Methodology, T.S., N.W., P.J., K.R.R. and T.O.; Resources, T.Z.O.; Supervision, K.R.R. and T.Z.O.; Validation, N.W., P.J. and T.Z.O.; Writing—original draft, T.S.; Writing—review and editing, N.W., P.J., K.R.R. and T.Z.O. All authors have read and agreed to the published version of the manuscript.

**Funding:** Estuarine Biogeochemistry Laboratory, Soil and Water Sciences Department, University of Florida.

**Acknowledgments:** Thank you to the Guana Tolomato Matanzas National Estuarine Research Reserve for use of sonde and meteorological data, as well as Trent Dye and Adam Pacetti for logistical support. We also appreciate Thomas Bianchi allowing use of his YSI data sonde during this storm event. We also greatly appreciate the availability and ability to use the U.S. Geological Survey hydrological data.

**Conflicts of Interest:** The authors declare no conflict of interest.

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
