# Peer review of "Impacts of Hurricane Disturbance on Water Quality across the Aquatic Continuum of a Blackwater River to Estuary Complex"

_jmse, doi:10.3390/jmse8060412_

Round 1

Reviewer 1 Report

The manuscript is well-presented and discussed; it focuses the dramatic impact of stormy event on aquatic estuarine system. I hope the authors, in future, will give information on biotic communities as stated in conclusion.

I have just to indicate a misprint

line 401 reference not according to journal guideline

and I have a curiosity in figure 8 (PCA), the mid-river and estuary points overlapping are the data during the storm event, correct?

Author Response

I have just to indicate a misprint

line 401 reference not according to journal guideline

Thank you for noticing this mistake. I have thoroughly combed through the manuscript and corrected the references to be in accordance with JMSE formatting guidelines.

and I have a curiosity in figure 8 (PCA), the mid-river and estuary points overlapping are the data during the storm event, correct?

Yes, during the storm event, salinity, DO, and turbidity were similar across the estuarine and mid-river sites. This is likely because the effects of the storm surge and precipitation increased all of these parameters temporarily. As the saltwater pushed into the river by the surge receded, salinity and DO across the entire system then decreased greatly. Due to the scale of these changes, the conditions at the estuarine and mid-river sites were very similar during (and for a couple of days after) the storm.

Reviewer 2 Report

The authors investigated the changes of water quality resulted from Hurricane events in aquatic ecosystem and its resilience and recovery. For this objective, they installed real-time system for collecting continuous time-varying data before, during, and after storm which allowed us to analyze the relationship between parameters. The authors concluded that the hurricane can affect the health of ecosystem. From this view point, the manuscript represents an interesting and useful input. Following concerns must be faced before being usable for publication.

  1. Line 139, 297, please provide effective photographs (FIGURE 1, FIGURE 8).
  2. Line 388, 451, citation correction
  3. In discussion section 4.2, Line 468-514, authors must provide more faithful representation regarding the ecosystem resilience and its environmental factors using this MS case study data.

Author Response

Line 139, 297, please provide effective photographs (FIGURE 1, FIGURE 8).

A photograph was added to figure one and a series of photographs have been added as a figure in the supplemental section (Fig. S1) in order to help the reader visualize the impacts on the study site.

Line 388, 451, citation correction

Thank you for directing us to this mistake. We have gone through the manuscript and worked to correct citation issues.

In discussion section 4.2, Line 468-514, authors must provide more faithful representation regarding the ecosystem resilience and its environmental factors using this MS case study data.

Thank you for the recommendation. We have provided additional discussion on our definition of resilience that clarifies the discussion of our data in the context of resilience (now line 491):

“A resilient system will return back to where it started after a disturbance relatively quickly whereas the end point of a less resilient system post disturbance would transition to an alternate “stable state” (i.e., not return to pre-storm conditions). This alternate stable state could be driven by landscape scale physical changes to the system (i.e., changes freshwater-marine exchange, changes to river width/drainage area, etc.) or biogeochemical alterations (i.e., changes in DOM inputs due to upstream impacts, changes in loading, etc.).”

Reviewer 3 Report

Presented manuscript deals with the issue of water quality changes in riverine and estuarine section of aquatic ecosystem as a result of severe disturbance caused by hurricane. Both, short- and long-term repercussions of strong winds and heavy precipitation were assessed in relation to the distance to the ocean shore. Variability in dissolved oxygen (DO) and organic matter (DOM) (and other water quality variables as well) was determined and explained to reveal the ecological resilience of different parts of creek ecosystem. I find this research very interesting and strongly broadening our knowledge in the matter of aquatic systems recovery after extremal disturbances.

The structure of the manuscript is acceptable, the figures clear and informative. Supplemental material allow for better understanding of the manuscript.

I suggest to publish the manuscript. I only have found some minor text editing errors:

Lines 41 and 44 – brackets?

Line 44 – coma after [9,10]

Line 50 – ‘and’ repetition

Line 199 – ‘are’ not necessary

Line 346 – shall it be ‘at last 15 times and were between’?

Line 480 – important?

Some of the references are not full – missing journal name or publisher, sometimes year, e.g. 1, 9, 10, 38, 47.

Author Response

Lines 41 and 44 – brackets?

Thank you for pointing out this editing mistake. The extra parentheses have been removed from the manuscript.

“Furthermore, numerous factors contribute to an ecosystems recovery and resilience including storm characteristics (angle and impact, intensity, surge, etc.) and in-situ­ characteristics such as nutrient reserves, microbial dynamics, biotic controls and ecosystem composition [9,10].

Line 44 – coma after [9,10]

Thank you for bringing this to our attention as well. The punctuation in this sentence has been corrected.

“ecosystem composition [9,10].”

Line 50 – ‘and’ repetition

The repetitive ‘and’s have been removed and replaced with more concise language.

“As biological communities, as well as terrestrial and aquatic inputs, change along the length of a river, OM composition and nutrient abundance evolves [11].”

Line 199 – ‘are’ not necessary

This issue has been fixed.

Line 346 – shall it be ‘at last 15 times and were between’?

Thank you for directing us to this language issue, it has been addressed in the manuscript.

 The estuary and tidal creek DOC concentrations increased by at least 15 times and were between 15-20 mg L-1 DOC at this time period.

Line 480 – important?

Thank you, we have changed “import” to “important”

Some of the references are not full – missing journal name or publisher, sometimes year, e.g. 1, 9, 10, 38, 47.

Thank you for bringing this to our attention. We have checked over the references more carefully and have rewritten them to align with journal standards.